# Biological Activities of the Fruit Essential Oil, Fruit, and Root Extracts of *Ferula drudeana* Korovin, the Putative Anatolian Ecotype of the Silphion Plant [note 1]

**DOI:** 10.3390/plants12040830

**Published:** 2023-02-13

**Authors:** Fatma Tosun, Fatih Göger, Gökalp İşcan, Mine Kürkçüoğlu, Fadıl Kaan Kuran, Mahmut Miski

**Affiliations:** 1Department of Pharmacognosy, School of Pharmacy, İstanbul Medipol University, İstanbul 34083, Turkey; 2Department of Pharmaceutical Botany, Faculty of Pharmacy, Afyonkarahisar Health Sciences University, Afyonkarahisar 03030, Turkey; 3Department of Pharmacognosy, Faculty of Pharmacy, Anadolu University, Eskişehir 26470, Turkey; 4Department of Pharmacognosy, Faculty of Pharmacy, İstanbul University, İstanbul 34116, Turkey

**Keywords:** *Ferula drudeana*, GC/MS, HPLC–ABTS^•+^, essential oil, antibacterial, anticandidal, antioxidant

## Abstract

In the present study, preliminary phytochemical investigations were performed on the fruit essential oil and antioxidant-rich methanolic extracts of the fruits and roots of *Ferula drudeana*, the putative Anatolian ecotype of the Silphion plant, to corroborate its medicinal plant potential and identify its unique characteristics amongst other *Ferula* species. The essential oil from the fruits of the endemic species *Ferula drudeana* collected from Aksaray was analyzed by GC and GC/MS. The main components of the oil were determined as shyobunone (44.2%) and 6-epishyobunone (12.6%). The essential oil of the fruits and various solvent extracts of the fruits and roots of *F. drudeana* were evaluated for their antibacterial and anticandidal activity using microbroth dilution methods. The essential oil of the fruits, methanol, and methylene chloride extracts of the fruits and roots showed weak to moderate inhibitory activity against all tested microorganisms with MIC values of 78–2000 µg/mL. However, the petroleum ether extract of the roots showed remarkable inhibitory activity against *Candida krusei* and *Candida utilis* with MIC values of 19.5 and 9.75 µg/mL, respectively. Furthermore, all the samples were tested for their antioxidant activities using DPPH^•^ TLC spot testing, online HPLC–ABTS screening, and DPPH/ABTS radical scavenging activity assessment assays. Methanolic extracts of the fruits and roots showed strong antioxidant activity in both systems.

## 1. Introduction

The genus *Ferula* (Apiaceae) comprises more than 220 species [1] and is widespread throughout the Mediterranean and Central Asia. It represents 26 species and 15 endemics in the flora of Turkey [2,3,4]. *Ferula* species are traditionally used in folk medicine for the treatment of many disorders, such as gastrointestinal problems, diarrhea, intestinal parasites, ulcer, hypotension, neurological disorders, epilepsy, rheumatism, and diabetes, and also used as a sedative, antispasmodic, expectorant, anticonvulsant, and tonic aphrodisiac [5]. About two thousand years ago, Pedanius Dioscorides described five drugs obtained from *Ferula* species in the third book of De Materia Medica [6], namely, Narthex (*Ferula communis* L.), Sagapenon (*F. persica* Wild.), Chalbane (Galbanum, *F. gummosa* Boiss.), Ammoniakon (*F. tingitana* L. or *F. marmarica* Asch. & Taub.), and Silphion, which clearly illustrates the span of medicinal use of *Ferula* species in ancient times.

*Ferula drudeana* Korovin (Figure 1) is a rare endemic species growing in the Central Anatolia region of Turkey. This species is the only member of the subgenus Narthex (Falc.) Drude of *Ferula* genus in Turkey [7]. With its unique morphological features and extremely limited local distributions near former Greek villages in Central Anatolia, *F. drudeana* has been proposed as an Anatolian ecotype of the silphion plant [8]. The silphion plant was used for many medicinal and culinary purposes during ancient times in Mediterranean countries. Pliny the Elder declared that “it would be an endless task to enumerate all the uses to which laser (i.e., the resin of silphion plant) is put” [9]. Theophrastus of Eresus calls the fruits of the silphion plant phyllon (i.e., leaf-like) [10]. Unfortunately, the use of the phyllon name inadvertently created an ambiguous terminology in the literature. The fruits of *Ferula drudeana* (Figure 2), like the fruits of other *Ferula* species, is a schizocharpic fruit that splits into two mericarps during their maturity. The fruits of Apiaceae plants are widely accepted sources of medicinal drugs due to their high essential oil/resin content. In contrast, their leaves are rarely used as a source of medicinal drugs [11]. Probably due to the use of ambiguous phyllon names for the fruits of the silphion plant by Theophrastus, instead of the biologically active metabolite-rich fruits, some authors referred to the leaves of the silphion plant as a medicinally used part [12]. In contrast, the medicinal use of the fruits of the silphion plant was not mentioned in the literature. As part of our study investigating the biological activities of the secondary metabolites of *Ferula drudeana*, we herein report on the analyses of the terpenoid content of the essential oil of the fruits as well as the major phenolic compounds of the methanolic extracts of the fruits and roots.

## 2. Results and Discussion

### 2.1. Volatile Composition of the Fruit Essential Oil of Ferula drudeana

The fruits of *F. drudeana* were collected in July and air-dried, and the coarsely crushed fruits were subjected to hydrodistillation to obtain its essential oil. The essential oil yield of the fruits of *F. drudeana* was 3.8%. The essential oil of the fruits was analyzed by GC-GC/MS systems, and 28 compounds representing 89.1% of the essential oil were characterized. The results of the analysis are shown in Table 1 and Figure 3.

The main components of the essential oil were identified as shyobunone (**9**) (44.2%), 6-epi-shyobunone (**8**) (12.6%), epi-isoshyobunone (**11**) (9.8%), and β-pinene (**1**) (5.8%). Oxygenated sesquiterpenes (74.5%), sesquiterpene hydrocarbons (8.6%), and monoterpene hydrocarbons (6.0%) were the main groups present in the oil. Oxygenated sesquiterpenes were the most abundant among these groups representing 74.5%. Previously, the fruit essential oil of another population of *Ferula drudeana* was analyzed, and its major components were identified as *epi*-isoshyobunone (38%), shyobunone (25%), and 6-*epi*-shyobunone (6%) [20]. Recently, the presence of high level of shyobunone derivatives, namely, isoshyobunone (23.9%), *epi*-shyobunone (18.9%), and shyobunone (2.7%), were discovered in the essential oil of fresh leaves of *Siparuna guianensis* Aubl. (Siparunaceae), a well-known Amazonian medicinal plant. The essential oil of *S. guianensis* was shown to have strong cholinesterase inhibitory, anti-Alzheimer, and neuroprotective activities due to its content of shyobunone derivatives [21]. The presence of shyobunone isomers in the essential oil of the aerial parts of *Daucus carota* L. var. *carota* (Apiaceae) was also reported in small quantities, namely, shyobunone (1.3%) and 6-epi-shyobunone (0.5%) [22]. Shyobunone and its isomers *epi*-shyobunone and isoshyobunone were originally isolated from the essential oil of the rhizomes of *Acorus calamus* L. (Acoraceae) [23]. The essential oil of the rhizomes of *Acorus calamus*, also known as sweet flag oil, was found to contain shyobunone (1.5–13.3%), 6-*epi*-shyobunone (0.4–3.1%), isoshyobunone (0.1–0.5%), and epi-isoshyobunone (3.3–7.3%) [13,24].

The essential oil composition of *Ferula drudeana* is unique amongst the *Ferula* species. Unlike other investigated *Ferula* species growing in the Mediterranean, Middle East, and North African countries, it has a very high percentage of oxygenated sesquiterpene compounds [8]. The proportion of rare elemane sesquiterpene ketone compounds in the essential oil is 66.6%.

### 2.2. Antimicrobial Testing of the Fruit Essential Oil of Ferula drudeana

Hydrodistilled fruit essential oil of *Ferula drudeana* demonstrated weak antimicrobial effects against all tested pathogenic Gram (+) and Gram (−) bacterial strains and the Candida panel with MIC values of 500–2000 μg/mL using CLSI M7-A7 and M27-A2 reference microdilution broth methods. Methanol and methylene chloride extracts of the fruits and roots displayed weak to moderate inhibitory effects against all tested microorganisms at concentrations between 78 and 2500 μg/mL. However, the petroleum ether extract of the roots (R1) showed remarkable inhibitory effects on *Candida krusei* and *Candida utilis* with MIC values of 19.5 and 9.75 μg/mL, respectively (Table 2 and Table 3). Our results are similar to previous works [25,26,27,28,29,30,31,32,33,34] on different *Ferula* essential oils and extracts.

In contrast to the essential oil fraction, the presence of strong antifungal activity in the fruit and root petroleum ether extracts suggests that the antifungal activity must be due to nonvolatile component(s) of these extracts. Further studies are in progress to identify the most potent antifungal compound(s) of these extracts. *C. albicans*, *C. tropicalis*, and *C. glabrata* represent the most clinically isolated *Candida* species. In contrast, other species, such as *C. krusei*, *C. parapsilosis*, *C. guilliermondii*, and *C. kefyr*, have also been isolated and are thought to be less virulent. However, recent data indicate that >30% of nosocomial *Candida* infections are due to species other than *C. albicans,* and in recent years, there has been a significant increase in *C. krusei*, a human pathogen causing systemic and ocular infections [35]. Identifying potent anticandidal substances in the petroleum ether extracts of *F. drudeana* may provide a promising antifungal agent for the treatment of such infections.

### 2.3. Antioxidant Activity Determination

#### 2.3.1. Qualitative TLC Spot Testing Evaluation of the Antioxidant Activities of the Fruit Essential Oil, Fruit, and Root Extracts of *Ferula drudeana*

The methanolic extracts of the fruits and roots of *Ferula drudeana* Korovin showed strong antioxidant activity by DPPH^•^ reagent treatment on a TLC silica gel plate Appendix A [36,37]. Following the application of the fruit essential oil, fruit, and root extract solutions to a silica gel plate, the plate was sprayed with DPPH (2,2-diphenyl-1-picryl-hydrazylhydrate) solution. No antioxidant activity was observed in the petroleum ether and methylene chloride extracts of the fruits and roots. Only a slight discoloration was observed on the periphery of the spot where the essential oil was applied to the silica gel plate. Peripheral discoloration of essential oil is probably related to the microlevel distribution of the essential oil by blank dilution solvent on the silica gel plate at the application point. The strong discoloration of the methanolic extracts of the fruits and roots of *F. drudeana* indicates the presence of antioxidant compounds in these extracts.

#### 2.3.2. Online HPLC–ABTS^•+^ Identification of Major Antioxidant Compounds of the Methanolic Extracts of *Ferula drudeana*

The methanolic extracts of the fruits and roots of *F. drudeana* were subjected to online high-performance liquid chromatography (HPLC)—2,2′-azinobis(3-ethylbenzothiazoline-6-sulfonic acid) radical cation (ABTS^•+^)-based screening assay for the identification of phenolic antioxidants of the fruits and roots of *F. drudeana* [38]. HPLC elute of the methanolic extracts of the fruits and roots of *F. drudeana* was split into two lines. The elute of one line mixed with a stabilized solution of ABTS^•+^ reagent, so the formation of negative peaks, indicating the antioxidant activity of the corresponding compound peaks, were monitored by measuring the decrease in absorbance at 734 nm (Figure 4). The elute of the second line was subjected to electrospray ionization mass spectrometry (EIMS) to identify the phenolic compounds responsible for the antioxidant activity of the methanolic extracts of fruits and roots of *F. drudeana*.

#### 2.3.3. HPLC–MS/MS Analysis of the Methanolic Extract of the Fruits of *Ferula drudeana*

The methanolic extract of the fruits of *F. drudeana* was subjected to high-performance liquid chromatography–MS/MS analysis to identify the phenolic compounds responsible for the antioxidant activity (Figure 5).

The mass spectrum of compound **13** (Appendix A; Table 4) (Rt 9.3 min) displayed a pseudo molecular [M-H]^−^ ion at *m*/*z* 353 and a base peak ion at *m*/*z* 191 [quinic acid-H]^−^ due to the loss of caffeoyl moiety (i.e., *m*/*z* 162), indicating a caffeic acid ester of quinic acid structure for **13**. When the caffeoyl group residue esterified at the 3-OH position of quinic acid, the intensity of the caffeoyloxy moiety (i.e., [caffeic acid-H]^−^) at *m*/*z* 179 is more than the connection to the 5-OH position of quinic acid [39]. Thus, the structure of **13** was identified as 5-caffeoylquinic acid (i.e., chlorogenic acid). The mass spectrum of compound **14** (Rt 12.8 min) was similar to that of **13** except for a pseudo molecular [M-H]^−^ at *m*/*z* 337, indicating the presence of a p-coumaric acid ester in **14** instead of a caffeic acid ester [40]. Consequently, the structure of **14** was confirmed as 5-*p*-coumaroylquinic acid.

The mass spectra of compounds **15** (Rt 18.7 min), **16** (Rt 19.4 min) Appendix A, and **17** (Rt 21.8 min) displayed the pseudo molecular ion [M-H]^−^ at *m*/*z* 515 and characteristic ion at *m*/*z* 353 due to the loss of a caffeoyl moiety (i.e., *m*/*z* 162). All of these compounds showed the quinic acid fragment ion at *m*/*z* 191 (i.e., [quinic acid-H]^−^) following the loss of the second caffeoyl moiety. These data suggested that compounds **15**, **16,** and **17** were isomers of the dicaffeoylquinic acid derivatives. The presence of *m*/*z* 191 [quinic acid-H]^−^ as the base peak and the absence of *m*/*z* 173 fragment in the mass spectra of compounds **15** and **16**
Appendix A suggested that these compounds could be 1,3-, 1,5-, or 3,5-dicaffeoyl-quinic acid derivatives [40]. Furthermore, the presence of a strong *m*/*z* 179 ion (i.e., [caffeoyloxy-H]^−^) fragment vs. the lack of a weak *m*/*z* 335 ion fragment suggested that the structure of **15** should be 3,5-dicaffeoylquinic acid [40]. In contrast, the presence of *m*/*z* 179 ion and weak *m*/*z* 335 ion fragments in the mass spectrum of **16**
Appendix A confirmed its structure as 1,5-dicaffeoylquinic acid (i.e., cynarine) [41].

The mass spectrum of compound **17** showed the main fragment ion at *m*/*z* 173, indicating the presence of a 4-OH substituted quinic acid. Consequently, this compound could be 3,4-dicaffeoylquinic acid or 4,5-dicaffeoylquinic acid. As the mass spectrum of 3,4-dicaffeoylquinic acid has an additional fragment at *m*/*z* 335 [41] and this fragment was absent in the mass spectrum of **17**, its structure should be 4,5-dicaffeoylquinic acid.

So far, several caffeoylquinic acid derivatives have been reported from the polar extracts of other *Ferula* species [45,46,47,48,49].

The mass spectrum of compound **18**
Appendix A showed *m*/*z* 447 [M-H]^−^ pseudo molecular ion and *m*/*z* 285 (luteolin aglycone) base peak ion due to the loss of 162 (i.e., hexose) moiety, which confirmed the structure of **18** as luteolin-7-glucoside. Previously, luteolin-7-glucoside has also been detected in other *Ferula* species [42,50].

The mass spectrum of compound **19** showed *m*/*z* 431 [M-H]^−^ pseudo molecular ion and *m*/*z* 268 (apigenin aglycone) base peak ion, which suggested the structure of **19** was apigenin glucoside. Apigenin-7-glucoside (**19**) has previously been identified in other *Ferula* species [43]. The mass spectroscopic data of compound **20** displayed *m*/*z* 461 [M-H]^−^ pseudo molecular and *m*/*z* 299 (hispidulin aglycone) ions as well as *m*/*z* 446, 283, and 255 fragment ions, which are in agreement with previously reported data. Thus, the structure of compound **20** was identified as hispidulin-7-glucoside [44].

#### 2.3.4. Determination of the Antioxidant Potential of the Methanolic Extracts of *Ferula drudeana* by DPPH and ABTS Free Radical Scavenging Activity Assessment

The antioxidant potential of the methanolic extracts of the fruits and roots of *F. drudeana* as well as two of their major antioxidant compounds, namely, chlorogenic acid and luteolin 7-glucoside, were determined by DPPH and ABTS radical scavenging activity tests. The results are shown in Table 5.

The DPPH radical scavenging test results indicated that the methanolic fruit extract of *F. drudeana* was about 2.2 times more active than the root extract. Chlorogenic acid and luteolin-7-glucoside, two of the major antioxidant compounds of methanolic extracts, were more potent antioxidants than butylated hydroxytoluene (BHT) but were not as effective as gallic or ascorbic acids.

The results of the TEAC experiment (i.e., ABTS radical scavenging activity) suggested that the methanolic extract of the fruits of *F. drudeana* was approximately 1.6 times more effective than the root extract at 1 mg/mL concentration. However, at 0.1 mg/mL concentration, none of the methanolic extracts was active. The antioxidant activity of luteolin-7-glucoside was higher than both BHT and ascorbic acid at 0.1 mg/mL concentration but lower than gallic acid. In this assay, chlorogenic acid was found to be a more potent antioxidant than any of the standards used.

### 2.4. Biological Activities of the Ferula drudeana Metabolites

The biological activities of the main essential oil terpenoids and phenolic compounds of the methanolic extracts of *Ferula drudeana* are summarized in Table 6 below.

## 3. Materials and Methods

### 3.1. General Experimental Procedures

The GC–FID analyses were carried out with capillary GC using an Agilent 6890N GC system (Agilent, Santa Clara, CA, USA), and the GC/MS analyses were performed on an Agilent 5975 GC–MSD system (Agilent, Santa Clara, CA, USA). An HP-Innowax FSC column (60 m × 0.25 mm, 0.25 μm film thickness, Agilent, Wilmington, DE, USA) was used for the analyses. The HPLC chromatographic separations were carried out using Shimadzu LC 20 System (Shimadzu, Tokyo, Japan). The mass spectra were recorded with AB Sciex 3200 Q TRAP mass spectrometer (AB Sciex, Toronto, Canada). GL Science Inertsil ODS 250 × 4.6 mm, 5 μm i.d. particle size, analytical column (GL Sciences, Tokyo, Japan) was used for the HPLC analyses. The turbidity of the standardized microbial sample solutions was measured using McFarland densitometer (Biosan McFarland Densitometer, Model Den-1B, Riga, Latvia). Antioxidant activity absorbances were recorded with a Biotek microplate reader (BioTek, Winooski, Vermont, USA). Chlorogenic acid, luteolin 7-glucoside, gallic acid, butylated hydroxytoluene (BHT), and L-ascorbic acid were purchased from Sigma-Aldrich (St. Louis, MO, USA).

### 3.2. Material

The plant material was collected (07 July 2012) near Mount Hasan in Aksaray, Turkey. A voucher specimen identified by Prof. Dr. H. Duman (Gazi University, Ankara) was deposited in the Herbarium of Gazi University (GAZI Nr. 9898000001568).

### 3.3. Extraction

Essential oil of the fruits of *Ferula drudeana* Korovin was obtained by hydrodistillation as described in Section 2.1 and subjected to GC–GC/MS analyses and antimicrobial testing. Air-dried and coarsely powdered fruits and roots (each 20 g) of *F. drudeana* were extracted in a Soxhlet extractor successively with petroleum ether (600 mL, 8 h), methylene chloride (600 mL, 8 h), and methanol (600 mL, 8 h). Each extract was concentrated in vacuo to remove the extraction solvent using a rotary evaporator and subjected to antimicrobial and antioxidant activities testing. The extract yields were as follows: fruit petroleum ether extract (**F1**) 1.916 g, fruit methylene chloride extract (**F2**) 0.265 g, fruit methanol extract (**F3**) 1.252 g, root petroleum ether extract (**R1**) 2.268 g, root methylene chloride extract (**R2**) 0.299 g, and root methanol extract (**R3**) 1.224 g.

### 3.4. Gas Chromatography–Gas Chromatography/Mass Spectrometry Analyses of Ferula drudeana Essential Oil

The oil was analyzed by capillary GC and GC/MS using an Agilent GC–MSD system (Agilent Technologies Inc., Santa Clara, CA, USA).

The GC/MS analysis was carried out with an Agilent 5975 GC-MSD system. Innowax FSC column (60 m × 0.25 mm, 0.25 μm film thickness) was used with helium as carrier gas (0.8 mL/min). GC oven temperature was kept at 60 °C for 10 min and programmed to 220 °C at a rate of 4 °C/min and kept constant at 220 °C for 10 min and then programmed to 240 °C at a rate of 1 °C/min. The split ratio was adjusted to 40:1. The injector temperature was set at 250 °C. MS were taken at 70 eV. The mass range was from *m*/*z* 35 to 450.

The GC analysis was carried out with an Agilent 6890N GC system fitted with a FID detector set at a temperature of 300 °C. To obtain the same elution order with GC/MS, simultaneous autoinjection was carried out on a duplicate of the same column applying the same operational conditions. Relative percentages of the separated compounds were calculated from FID chromatograms.

The components of essential oils were identified by comparing their mass spectra with those in the Baser Library of Essential Oil Constituents, Wiley GC/MS Library, Adams Library, and Mass Finder Library and confirmed by comparison of their retention indices. Alkanes were used as reference points to calculate the relative retention indices (RRI). Relative percentages of the separated compounds were calculated from FID chromatograms. The results of the analysis are shown in Table 1.

### 3.5. Tested Microorganisms and Standard Antimicrobial Agents

*Escherichia coli* NRRL B-3008, *Pseudomonas aeruginosa* ATCC 27853, *Salmonella typhimurium* ATCC 13311, *Bacillus cereus* NRRL B-3711, *B. subtilis* NRRL B-4378, *Serratia marcescens* NRRL B-2544, *Staphylococcus epidermidis* ATCC 12228, *E. coli* O157:H7 RSSK 234 (RSSK; **RSHM National Type Culture Collection Strains of Bacteria**), two different strains of *Candida albicans* (clinically isolated, Osmangazi University, Faculty of Medicine, Department of Microbiology and ATCC 90028), *C. utilis* NRRL Y-12968, *C. krusei* NRRL Y-7179, and *C. glabrata* (clinically isolated, Osmangazi University, Faculty of Medicine, Department of Microbiology and ATCC 90028) were used as the test microorganisms. Chloramphenicol (Merck, Rahway, NJ, USA), ampicillin (Merck), amphotericin-B (Sigma-Aldrich), and ketoconazole (Sigma-Aldrich) were used as standard antimicrobial agents.

### 3.6. Antimicrobial Activity

Antibacterial and antifungal activities of the samples were evaluated using slightly modified CLSI (formerly NCCLS) microdilution broth methods M7-A7 and M27-A2, respectively [160,161].

### 3.7. Antioxidant Activity

The antioxidant activity of *Ferula drudeana* essential oil and extracts were evaluated by DPPH^•^ TLC spot testing assay, the major antioxidant compounds of the active extracts were identified by online HPLC–ABTS^•+^ screening assay coupled with HPLC–UV–MS/MS system, and the antioxidant potential of methanolic extracts of *F. drudeana* was determined with DPPH and ABTS radical scavenging activity testing.

#### 3.7.1. DPPH^•^ TLC Spot Testing

The DPPH^•^ TLC spot testing method was used to find the active antioxidant extracts. The extract and essential oil solutions (10 μL of 1 mg/mL) of *Ferula drudeana* were spotted on a silica gel TLC plate, then 2.54 mM DPPH–methanol solution was sprayed using a Camag TLC sprayer, and the results were evaluated after 30 min. Spots with the DPPH solution scavenging activity were observed as white-yellow spots on a purple background Appendix A [36,37].

#### 3.7.2. HPLC–ABTS^•+^ Derivatization

Online HPLC–ABTS screening and HPLC–UV–MS/MS method were used, and the extract’s active compounds were identified using the method developed by Koleva and He [30,162]. HPLC coupled with ABTS assay was performed using a stock solution containing 3.5 mM potassium persulphate, and 2 mM ABTS was prepared and kept at room temperature in darkness for 16 h to stabilize the radical. The radical reagent was prepared by diluting the stock solution with pure water to an absorbance of 0.70 ± 0.02 at 734 nm. The extracts (at 10 mg/mL concentration, 10 µL) were injected into a Shimadzu HPLC system. HPLC separation was carried out as described in the previous section. HPLC analytes from the column arrived at a T-junction, where the ABTS reagent was added. A Shimadzu reagent pump delivered the ABTS reagent at a 0.7 mL/min flow rate. After the analytes were mixed with ABTS reagent in a reaction coil (15 m–0.25 mm i.d. PEEK tubing), DAD measured the negative peaks at 734 nm. Data were analyzed using LC Solution Software.

#### 3.7.3. LC–MS/MS Analysis

AbSciex 3200 MS/MS detector was used for LC–MS/MS analysis. A negative ionization mode was preferred for ionization. Chromatographic separations were carried out with GL science Inertsil ODS 250 × 4.6 mm, i.d., 5 µm column using Shimadzu 20A HPLC. The column oven temperature was set to 40 °C, and the flow rate was adjusted to 1 mL/minute. Mobile phases were (A) methanol: water: formic acid (10:89:1, *v*/*v*/*v*) and (B) methanol: water: formic acid (89:10:1, *v*/*v*/*v*). The B concentration increased from 15% to 40% in 15 min, then increased to 45% within 3 min; it was at 45% B at 12 min, then increased to 75% within 5 min and 100% within 5 min. For mass scanning (EMS), a mass range of 100–1000 amu was chosen.

#### 3.7.4. Determination of the Antioxidant Potential of Methanolic Extracts

##### DPPH Radical Scavenging Activity

Using a modified version of the Brand-Williams method [163,164], the DPPH radical scavenging capacity of the fruit and root methanolic extracts, standards, and positive control standards were assessed.

Solutions of the methanolic extracts of fruits and roots of *F. drudeana* (1.25 mg/mL), two standard compounds (0.1 mg/mL of each luteolin-7-glucoside and chlorogenic acid), and three positive control standards (0.1 mg/mL of each gallic acid and ascorbic acid and 1 mg/mL BHT) in methanol were prepared. In 96-well flat-bottom plates, 100 μL of the sample solutions (extracts, standards, and positive control standards) were serially diluted with 100 μL of methanol. Then, diluted samples were mixed with 100 μL of DPPH solution (0.08 mg/mL in methanol). As a control, 100 μL of methanol and 100 μL of DPPH solution were combined. The mixtures were kept in the dark for 30 min. Absorbance of each well was recorded at 517 nm. The percentage of inhibition was calculated using the following equation:% Inhibition = [(A_control_ − A_sample_)/A_control_] × 100
where A_control_ is the absorbance of the solution that contains all reagents with the exception of extract or standard chemical. The 50% inhibition concentration (IC_50_) values of DPPH radical of each sample were calculated using SigmaPlot (Version 12.0).

##### ABTS Radical Scavenging Activity

ABTS radical scavenging test was performed following the method described by Re et al. [165] with slight modification. In order to produce the ABTS^•+^ free radical cation, 7 mM ABTS and 2.5 mM K_2_S_2_O_8_ were dissolved in 10 mL ultrapure water and incubated at room temperature for 16 h in the dark. The ABTS^•+^ solution was diluted with 100% ethanol prior to the experiment to obtain an absorbance of 0.7–0.8 at 734 nm. Solutions were prepared for extracts (0.1–1 mg/mL), standards (0.1–1 mg/mL), and Trolox (3.0, 2.0, 1.0, 0.5, 0.25, and 0.125 mM). During the experiment, 990 μL of ABTS^•+^ solution was combined with 10 μL of the sample (extract/standard) solution, and the combined mixtures were allowed to incubate for 30 min at room temperature in the dark. Trolox equivalent antioxidant capacity (TEAC) of the extracts and standards were calculated using the linear equation obtained for Trolox (y = 20.227x + 4.1663 R^2^_ = 0.9968).

## 4. Conclusions

This is the first report on the bioactivities of terpenoids present in the essential oil and major phenolic compounds of methanolic extracts of *Ferula drudeana*. The fruit essential oil of *F. drudeana* was analyzed by GC–MS, and 28 terpenoid compounds were identified. About 75% of the essential oil consisted of oxygenated sesquiterpene compounds. The presence of unusually high levels of shyobunone derivatives with reported anti-Alzheimer and neuroprotective activities in the essential oil of *F. drudeana* clearly highlights the medicinal value of this species. Although the antimicrobial activity of the fruit essential oil of *F. drudeana* was weak to moderate, the presence of strong anticandidal activity in the fruit and root petroleum ether extracts suggested the source of anticandidal activity was probably the nonvolatile component(s) of this extract. The results of DPPH^•·^ TLC spot testing, online HPLC–ABTS^•+^ screening assay, and DPPH/ABTS radical scavenging tests indicated that *F. drudeana* was rich in polyphenolic compounds and exhibited good antioxidant activity. Three flavonoid glucosides and five hydroxycoumaric acid esters of quinic acid were determined in the methanol extracts of the fruits and roots of *F. drudeana* by HPLC–MS/MS analyses, and their antioxidant activities were detected simultaneously by online coupled HPLC–ABTS^•+^ based assay. Furthermore, based on the literature survey, known biological activities of the major essential oil terpenoids and methanolic extract phenolic compounds of *F. drudeana* were reviewed to confirm the medicinal values of *F. drudeana* secondary metabolites.

## Figures and Tables

**Figure 1 plants-12-00830-f001:**
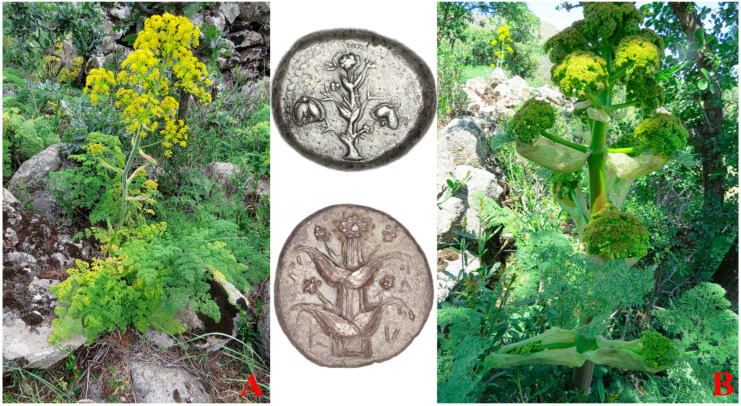
(**A**) The general view of *Ferula drudeana* is similar to the numismatic silphion plant figures on the early period Cyrenaic coins (upper coin figure). (**B**) The numismatic figures on the later period Cyrenaic coins (lower coin figure) resemble the developing premature flowering stem of *F. drudeana*. Copyrights of coin figures: Trustees of the British Museum.

**Figure 2 plants-12-00830-f002:**
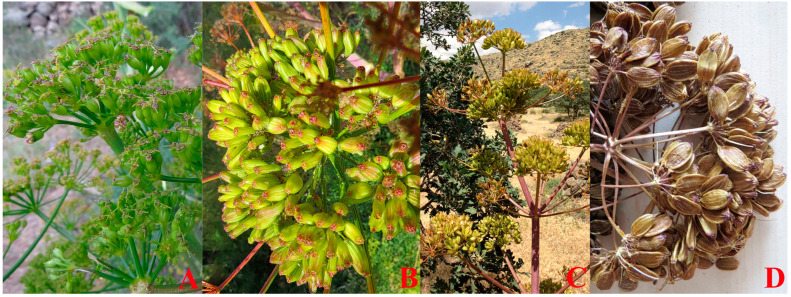
Fruits of *Ferula drudeana* during their development stages: (**A**) immature fruits, (**B**) young fruits, (**C**) maturing young fruits become yellow due to the loss of chlorophyll, (**D**) golden brown mature fruits.

**Figure 3 plants-12-00830-f003:**
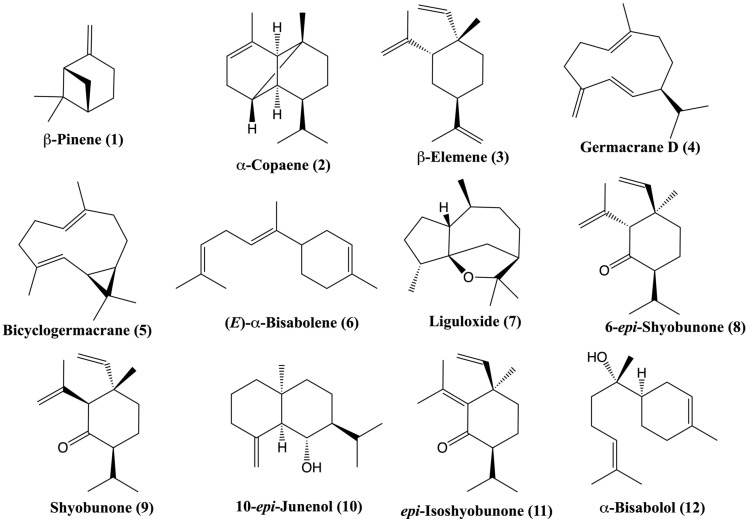
Main terpenoid compounds of the fruit essential oil of *Ferula drudeana*.

**Figure 4 plants-12-00830-f004:**
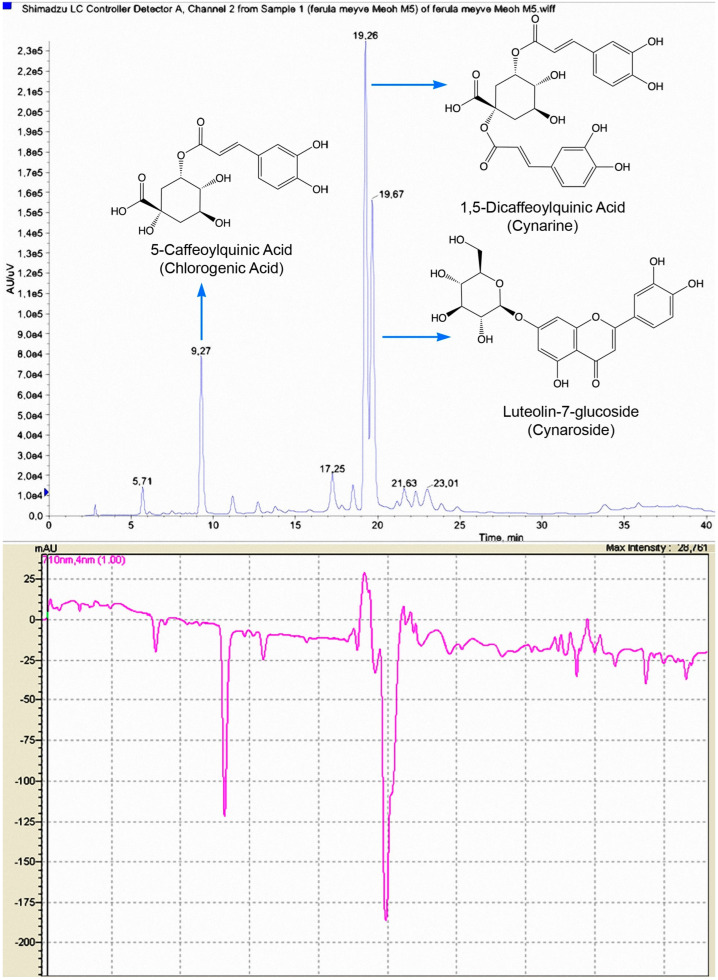
Online HPLC–ABTS^•+^ chromatogram (lower section) of the methanolic extract of fruits of *Ferula drudeana* display negative peaks, indicating the antioxidant activity of the corresponding compound peaks in the upper HPLC chromatogram.

**Figure 5 plants-12-00830-f005:**
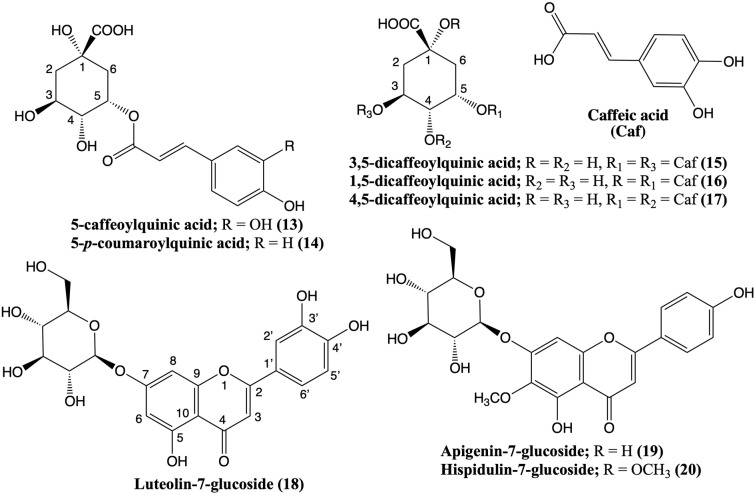
Structures of the antioxidant phenolic compounds of the fruit and root methanolic extracts of *Ferula drudeana*.

**Table 1 plants-12-00830-t001:** Volatile composition of the fruit essential oil of *Ferula drudeana*.

No.	RRI ^a^	RRI ^b^	Molecular Formula/MW	Compound Name	Peak Area (%)	IM
1	1032	1032 ^c^, 1008–1039 ^d^	C_10_H_16_/136	α-Pinene	0.2	t_R_, MS
2	1118	1118 ^c^, 1085–1130 ^d^	C_10_H_16_/136	β-Pinene (**1**)	5.8	t_R_, MS
3	1132	1132 ^c^	C_10_H_16_/136	Sabinene	tr	t_R_, MS
4	1174	1174 ^c^	C_10_H_16_/136	Myrcene	tr	t_R_, MS
5	1299	1299 ^e^	C_10_H_20_O_2_/172	2-Methylbutyl isovalerate	tr	MS
6	1491	1495 ^c^	C_15_H_24_/204	Bicycloelemene	0.1	MS
7	1492	1445–1549 ^d^	C_15_H_24_/204	Cyclosativene	0.4	MS
8	1497	1497 ^c^	C_15_H_24_/204	α-Copaene (**2**)	0.5	MS
9	1550	1559 ^d^, 1534–1580 ^d^	C_15_H_24_/204	*cis*-α-Bergamotene	0.2	MS
10	1597	1583–1668 ^d^	C_15_H_24_/204	α-Guaiene	0.3	MS
11	1600	1565–1608 ^d^	C_15_H_24_/204	β-Elemene (**3**)	0.6	MS
12	1612	1612 ^c^	C_15_H_24_/204	β-Caryophyllene	0.2	t_R_, MS
13	1621	-	-/204	Unknown 1	1.8	MS
14	1668	1669 ^e^	C_15_H_24_/204	(*Z*)-β-Farnesene	0.2	MS
15	1669	1668 ^c^, 1627–1668 ^d^	C_15_H_24_/204	Sesquisabinene	0.3	t_R_, MS
16	1671	1643–1684 ^d^	C_15_H_24_/204	(*E*)-β-Farnesene	0.1	MS
17	1687	1687 ^c^	C_15_H_24_/204	α-Humulene	0.1	t_R_, MS
18	1719	1718 ^f^	C_15_H_24_/204	γ-Guaiene	0.2	MS
19	1726	1726 ^c^	C_15_H_24_/204	Germacrene D (**4**)	1.5	MS
20	1755	1755 ^c^	C_15_H_24_/204	Bicyclogermacrene (**5**)	2.7	t_R_, MS
21	1772	1773 ^c^	C_15_H_24_/204	δ-Cadinene	0.2	t_R_, MS
22	1784	1773–1786 ^d^	C_15_H_24_/204	(*E*)-α-Bisabolene (**6**)	1.0	MS
23	1804	1812 ^g^, 1808 ^h^	C_15_H_26_O/222	Liguloxide (**7**)	1.6	MS
24	1868	1861 ^c^	C_15_H_24_O/220	6-*epi*-Shyobunone (**8**)	12.6	MS
25	1900	1893 ^c^	C_15_H_24_O/220	Isoshyobunone	tr	MS
26	1916	1903 ^c^	C_15_H_24_O/220	Shyobunone (**9**)	44.2	MS
27	1977	2028 ^k^, 2052 ^k^	C_15_H_26_O/222	10-*epi*-Junenol (**10**)	5.8	MS
28	2053	2044 ^c^	C_15_H_24_O/220	*epi*-Isoshyobunone (**11**)	9.8	MS
29	2084	-	-/236	Unknown 2	2.2	MS
30	2092	-	-/220	Unknown 3	1.1	MS
31	2232	2178–2234 ^d^	C_15_H_26_O/222	α-Bisabolol (**12**)	0.5	t_R_, MS
				Monoterpene Hydrocarbons	6.0	
				Sesquiterpene Hydrocarbons	8.6	
				Oxygenated Sesquiterpenes	74.5	
				Others	5.1	
				Total %	94.2	

Compounds listed in order of their elution in HP Innowax FSC GC column. RRI ^a^: relative retention indices experimentally calculated against n-alkanes; RRI ^b^: RRI from literature (^c^ [13], ^d^ [14], ^e^ [15], ^f^ [16], ^g^ [17], ^h^ [18], and ^k^ [19]) for polar column values, with % calculated from FID data; tr: trace (<0.1%); IM: identification method; t_R_: identification based on comparison with coinjected standards on an HP Innowax column; MS: identification based on computer matching of the mass spectra libraries.

**Table 2 plants-12-00830-t002:** Antibacterial activity of the fruit essential oil, fruit, and root extracts of *Ferula drudeana* (MIC, μg/mL).

Microorganisms	EOF	F1	F2	F3	R1	R2	R3	S1	S2
*Escherichia coli*	2000	1250	1250	2500	1250	1800	1800	3.9	1
*Pseudomonas aeruginosa*	2000	625	1250	2500	1250	900	900	62.5	15.6
*Salmonella typhimurium*	500	1250	1250	2500	625	900	900	3.9	1
*Bacillus cereus*	1000	2500	625	1250	1250	3600	450	7.8	1
*Bacillus subtilis*	1000	1250	1250	2500	1250	450	900	1.9	1
*Serratia marcescens*	1000	625	625	1250	1250	450	900	15.6	15.6
*Staphylococcus epidermidis*	2000	2500	625	625	312	1800	900	3.9	1
*E. coli* O157:H7	2000	1250	1250	2500	625	900	1800	3.9	1

**EOF**: essential oil of the fruits; **F**: fruit extracts; **R**: root extracts (1: petroleum ether, 2: methylene chloride, and 3: methanol); **S1**: chloramphenicol; **S2**: ampicillin.

**Table 3 plants-12-00830-t003:** Anticandidal activity of the fruit essential oil, fruit, and root extracts of *Ferula drudeana* (MIC, μg/mL).

Microorganisms	EOF	F1	F2	F3	R1	R2	R3	S1	S2
*Candida albicans* *	250	312	1250	1250	156	450	900	0.05	0.1
*Candida utilis*	500	39	78	312	19.5	112	225	1.6	0.05
*Candida tropicalis*	2000	625	312	1250	310	900	450	0.2	0.2
*Candida krusei*	500	39	312	625	9.75	450	900	1.6	0.2
*Candida albicans*	2000	312	1250	1250	156	450	450	0.1	0.2
*Candida glabrata*	2000	156	625	156	78	225	450	3.2	0.2

**EOF**: essential oil of the fruits; **F**: fruit extracts; **R**: root extracts (1: petroleum ether, 2: methylene chloride, and 3: methanol); **S1**: ketoconazole; **S2**: amphotericin-B; *: clinically isolated strain.

**Table 4 plants-12-00830-t004:** Peak assignment for the HPLC–MS/MS analysis of the methanol extracts of *Ferula drudeana*.

No	Rt (min)	[M-H]^−^ (*m*/*z*)	Fragment Ion (*m*/*z*)	Identification	Reference
1	9.3	353	353 (16), 191 (100), 179 (5)	5-Caffeoylquinic acid (13)	[40,41]
2	11.4	293	293 (100), 131 (20)	Unknown 4	
3	12.8	337	337 (16), 191 (100)	5-*p*-Coumaroylquinic acid (14)	[40]
4	17.4	515	191 (31), 179 (31), 173 (46), 135 (100)	Unknown 5	
5	18.7	515	515 (43), 353 (43) 191 (100) 179 (38)	3,5-Dicaffeoylquinic acid (15)	[40]
6	19.4	515	515 (23), 353 (35), 335 (5), 191 (100), 179 (12), 161 (9)	1,5-Dicaffeoylquinic acid (16)	[41]
7	19.7	447	447 (73), 285 (100)	Luteolin-7-glucoside (18)	[42]
8	21.8	515	515 (17), 353 (17), 191 (39), 179 (44), 173 (100)	4,5-Dicaffeoylquinic acid (17)	[41]
9	23.1	431	431 (84), 268 (100)	Apigenin-7-glucoside (19)	[43]
10	23.9	461	461 (100), 446 (32), 313 (11), 298 (42), 283 (37), 255 (68)	Hispidulin-7-glucoside (20)	[44]

**Table 5 plants-12-00830-t005:** Antioxidant activities of *Ferula drudeana* extracts and standards.

Extracts/Compounds	DPPH [IC_50_, mg/mL] ^2^	TEAC ^1^ [mM] 1 mg/mL	TEAC [mM] 0.1 mg/mL
Fruit Methanolic Extract of *F. drudeana*	0.087 ± 0.011	0.41 ± 0.07	na ^3^
Root Methanolic Extract of *F. drudeana*	0.189 ± 0.048	0.25 ± 0.09	na
Chlorogenic acid ^4^	0.013 ± 0.001	2.94 ± 0.29	0.16 ± 0.07
Luteolin-7-glucoside ^4^	0.0073 ± 0.0005	3.02 ± 0.05	1.03 ± 0.02
Gallic acid ^5^	0.002 ± 0.0001	3.22 ± 0.02	3.24 ± 0.01
BHT ^5^	0.042 ± 0.008	3.16 ± 0.04	0.37 ± 0.05
Ascorbic acid ^5^	0.006 ± 0.001	3.24 ± 0.05	0.73 ± 0.07

^1^ TEAC: Trolox equivalent antioxidant capacity; ^2^ IC_50_: 50% inhibition concentration; ^3^ na: not active; ^4^ standard; ^5^ positive control standard.

**Table 6 plants-12-00830-t006:** Biological activities of the compounds identified in the fruit essential oil, fruit, and root methanol extracts of *Ferula drudeana*.

Secondary Metabolite	Biological Activities
β-Pinene (**1**)	Antibacterial, anticandidal [51,52], antibiofilm [53], phytotoxic [54], antidepressant-like activity [55], cytotoxic [56], gastroprotective [57], anticonvulsant [58]
α-Copaene (**2**)	Cytotoxic, antioxidant, antigenotoxic [59,60], insect attractant [61], analgesic and anti-inflammatory [62]
β-Elemene (**3**)	Antitumor, anticancer activity [63,64,65,66,67,68], antimigraine [69]
Germacrene D (**4**)	Cytotoxic, antioxidant, insecticidal [70], insect attractant [71], antibacterial [72]
Bicyclogermacrene (**5**)	Larvicidal [73], radical scavenger [74]
(*E*)-α-Bisabolene (**6**)	Antioxidant [75], cytotoxic [76], anti-inflammatory [77], antifungal [78]
Shyobunone (**8**)	Insecticidal, repellent activity [79], neuroprotective, cholinesterase inhibitor, anti-Alzheimer [21], antibacterial (against *Helicobacter pylori*) [80]
6-*epi*-shyobunone (**9**)	Cholinesterase inhibitor, anti-Alzheimer, neuroprotective [21]
*Epi*-isoshyobunone (**11**)	Insecticidal, repellent activity [79], cholinesterase inhibitor, anti-Alzheimer, neuroprotective [21]
α-Bisabolol (**12**)	Anti-inflammatory, analgesic [81], antioxidant, anti-infective [82], cytotoxic [83], gastroprotective [84], nephroprotective [85], uterorelaxant [86], antileishmanial [87], antitumoral [88]
Cynarine (**16**) (1,5-Dicaffeoylquinic acid)	Antimicrobial [89], hepatoprotective [90], antihypertensive, vasodilator [91], choleretic [92], antioxidant [93], anti-inflammatory [94,95], antidiabetic [96,97], antidepressant [98], antivitiligo [99], anti-HIV-1 [100], Ebola virus inhibitor [101], Janus kinase (JAK) inhibitor [102]
Chlorogenic acid (**13**) (5-Caffeoylquinic acid)	Antimicrobial [89], hepatoprotective [90], antihypertensive, vasodilator [91], antitumor [103], anti-inflammatory [104], improves late diabetes [105], protects against cholestatic liver injury [106], neuroprotective [107], antiviral activity against influenza A (H1N1/H3N2) virus [108], antidiabetic and antilipidemic [109], inhibits hepatocellular carcinoma [110], anxiolytic and antioxidant [111], antihyperalgesic [112], cardioprotective [113], neuroprotective and cognitive improvement [114], improves hepatic steatosis and insulin resistance [115], alleviates obesity and modulates gut microbiota [116], ameliorates ulcerative colitis [117], inhibits glioblastoma growth [118], induces 4T1 breast cancer tumor’s apoptosis [119], strong matrix metalloproteinase-9 inhibitor [120]
3,5-Dicaffeoylquinic acid (**15**) (Isochlorogenic acid A)	Promotes melanin synthesis [121], antirosacea [122], antioxidant [93,123]
4,5-Dicaffeoylquinic acid (**17**) (Isochlorogenic acid C)	Antirosacea [122], antioxidant [93,123]
Cynaroside (**18**) (Luteolin-7-glucoside)	Choleretic and anticholestatic [124], antioxidant [125,126,127,128], anticholinesterase [125], antibacterial against multidrug-resistant clinical isolate strains [129], anti-inflammatory [128,130,131,132], antiallergic [132], inhibitor of monoamine oxidase B [133], inhibitor of low-density lipoprotein oxidation [134], antidiabetic [135,136,137], antidepressant [138], cytotoxic, anticancer [139,140,141,142,143,144], antimicrobial [89,145], antimutagenic [145], hepatoprotective [90], chondroprotective [146], CYP1A2 inhibitor [147], intestinal motility [148], retinal protective [149]
Apigenin-7-glucoside (**19**)	Antibacterial [150], antibacterial and antifungal [151], inhibition of α-amylase activity [152], anticandidal [153], cytotoxic, anticancer [142,153,154,155]
Homoplantaginin (**20**) (Hispidulin- 7-glucoside)	Antioxidant [156,157], antiproteasomal [158], collagenase, elastase and hyaluronidase enzyme inhibitory [159]

## Data Availability

The Appendix A for the DPPH^•^ TLC spot testing and LC-MS/MS analyses of chloro-genic acid, cynarine and cynaroside are available online.

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
