# Peer review of "Biological Activities of the Fruit Essential Oil, Fruit, and Root Extracts of Ferula drudeana Korovin, the Putative Anatolian Ecotype of the Silphion Plant†"

_plants, 2023, doi:10.3390/plants12040830_

Round 1
Reviewer 1 Report
In the scientific article "Biological Activities of the Fruit Essential Oil, Fruit and Root 2
Extracts of Ferula drudeana Korovin, Putative Anatolian Eco- 3
type of the Silphion Plant," antibacterial, anticandidal, antioxidant analyses of the essential oils I esctractracts of the plant named in the title were performed. In addition, their GC/MS and HPLC-ABTS+ spectra are presented. The article is structured correctly, the subject matter is comprehensibly presented. The paper uses modern analytical methods . However, I have one big reservation about the methodology and chapter of The DPPH- TLC Bioautographic Determination. I also analyzed the article the authors refer to in the reference for this method. The results presented in that article are completely different from those presented in this paper. In your paper, I only came across images that can illustrate to some extent whether a given extract/oil has potential antioxidant properties. On the other hand, in the paper you cite there are results given in % radical scavenging and expressed mg/g equivalent. I would expect such values in your paper as well, so that the antioxidant potential can be accurately confirmed. From your antioxidant results, I can only guess that the extracts/oils have some antioxidant potential, and reading papers in such a respected journal, I need to be sure at what level exactly this potential is. This is my only and most significant comment of this paper.
The paper is suitable for publication in significant correction of DPPH results.

Author Response
Dear Reviewer 1,
Thank you very much for your invaluable comments about our manuscript; we have extensively revised our manuscript to address your comments.
We agree that the section title of "DPPH TLC Bioautographic Determination" was misleading. This test was not performed for a bioautographic determination of potential antioxidant components of Ferula drudeana extracts but to identify the F. drudeana extracts that contain antioxidant compounds. Thus, the section title of that experiment was revised to "Qualitative TLC spot testing evaluation..." to clarify the non-bioautographic nature of the test. Following the identification of antioxidant compounds containing extracts of F. drudeana, the antioxidant components of those extracts were determined by online HPLC-ABTS•+ procedure followed by the GC-MS/MS identification of individual antioxidant compounds of F. drudeana methanolic extracts. In addition, the antioxidant potential of the methanolic extracts of F. drudeana (and standards of the two of the identified antioxidant compound) was determined using DPPH and ABTS radical scavenging activity tests per your suggestions.
We hope that these additional studies provide satisfactory answers to your comments.
Kind regards,
M. Miski, Ph.D.
Reviewer 2 Report
The authors in this manuscript present information about biological activity and compounds from extracts Ferula drudeana Korovin. This study is quite interesting. The authors conducted and presented several important analyzes, such as GC/MS, LC-MS/MS, antioxidant and antimicrobial activity.
My general comments are presented below
Figures 2, 5, 6, 7 are not necessary and should be move to supplementary material.
Improve quality of figure 3.
Could a quantitative analysis also have been done, only a qualitative one was presented?
Introduction
I suggest to add some information about bioactive compounds and activity of Ferula extract, based on previous literature. This aspect was omitted in introduction.
2.3.1. The DPPH• TLC Bioautographic Determination
Please compare your results with other findings. Is it strong or weak antioxidant activity? In other plants non polar fraction could be responsible for antioxidant activity?
3.2. Extraction
Provide more detailed information about extraction procedures. It was crucial step for other analysis. What was mass of sample, sample:solvent ratio, extraction temperature, etc.
Detailed comments are presented below
Line 102 ‘(%80)’ mistyping?
Line 119 It was fresh fruit? Provide more detailed information, temperature, time of extraction, ratio.
Line 220 What was air-drying conditions? It was the same for fruit and roots?
Line 257-258 What does it mean ‘partly 258 modified method’?
Line 293 ‘v/v/h’ mistyping?
Line 298 rather ‘present’ then ‘content’. Where did you calculated amount of terpenoids?
Table 2, 3 and Figure 2, maybe introduce sample codes in the materials and methods section. You can remove long and unclear footnotes.
Figures 5-7 are not necessary, they duplicate the information from Table 4. Rather, make references to Table 4 in the text.
Author Response
Dear Reviewer 2,
Thank you very much for your crucial comments.
My general comments are presented below
Figures 2, 5, 6, 7 are not necessary and should be move to supplementary material.
Those figures were transferred into the supplementary material of original manuscript.
Improve quality of figure 3.
The quality of figure 3 (i.e., figure 4 in the revised manuscript) was improved.
Could a quantitative analysis also have been done, only a qualitative one was presented?
A quantitative assessment of antioxidant analyses was also performed and included in the revised manuscript in section 2.3.4.
Introduction
I suggest to add some information about bioactive compounds and activity of Ferula extract, based on previous literature. This aspect was omitted in introduction.
Additional information about the medicinal value, ethnobotanical uses, and historical importance of Ferula species was included in the introduction section of the revised manuscript.
2.3.1. The DPPH• TLC Bioautographic Determination
Please compare your results with other findings. Is it strong or weak antioxidant activity? In other plants non polar fraction could be responsible for antioxidant activity?
This section was completely revised to explain the actual intended purpose of this test; it was used to identify the extract of F. drudeana with the most antioxidant activity so that we could determine the secondary metabolites with strong antioxidant activity and assess their antioxidant potential with various radical scavenging tests (please see the section 2.3.4 in the revised manuscript).
3.2. Extraction
Provide more detailed information about extraction procedures. It was crucial step for other analysis. What was mass of sample, sample:solvent ratio, extraction temperature, etc.
Detailed extraction information is provided in section 3.3 of the revised manuscript.
Detailed comments are presented below
Line 102 ‘(%80)’ mistyping?
(%80) is removed in the revised manuscript.
Line 119 It was fresh fruit? Provide more detailed information, temperature, time of extraction, ratio.
Line 220 What was air-drying conditions? It was the same for fruit and roots?
Air-dried (in a well-ventilated shady area at room temperature) and coarsely crushed fruits were extracted, and the drying condition for the roots was similar. The detailed extraction conditions were provided in section 3.3 of the revised manuscript.
Line 257-258 What does it mean ‘partly 258 modified method’?
The only difference between the literature method vs. our method was "the stock solutions of the plant essential oil and extract samples were prepared in higher concentrations than the standard antimicrobial agent concentrations (i.e., positive reference standards)."
Line 293 ‘v/v/h’ mistyping?
It was a typographic error and corrected in the revised manuscript.
Line 298 rather ‘present’ then ‘content’. Where did you calculated amount of terpenoids?
This section is corrected in the revised manuscript; the procedure for the calculation of the amounts of terpenoids is described in sections 2.1 and 3.4.
Table 2, 3 and Figure 2, maybe introduce sample codes in the materials and methods section. You can remove long and unclear footnotes.
Figures 5-7 are not necessary, they duplicate the information from Table 4. Rather, make references to Table 4 in the text.
All the required corrections were implemented in the revised manuscript.
Thank you very much for your suggestions and comments.
Kind regards,
M. Miski, Ph.D.
Reviewer 3 Report
The revision is attached in a separate document.

Author Response
Dear Reviewer 3,
Thank you very much for your precious comments and question, which allowed us to improve our manuscript.
General Comments:
1. First weakness refers to the hypothesis which is not sufficiently explained. Please supply more information which explain why is worth to analyse the plant from genus Ferula regarding pharmaceutical, nutritional (and other crucial properties).
Additional information was included in the abstract and introduction section of the revised manuscript to address this comment.
2. I suggest to insert in Introduction or Materials and Methods part photo of analysed plant which is not commonly known. Also please consider and indicate which part of plant composition was analysed fruits or seeds (it is confusing).
Pictures of the general view of Ferula drudeana and its fruits were included in the introduction section of the revised manuscript; parts of the plant analyzed in the present study were the fruits and roots.
3. It is not clear which morphological part of plant were analysed material. From which part extract were obtained. Which solvent was used ? Please consider to include chart or Table for presenting experimental material (part 3). Also please supply this information in caption of used methods (Part 3) and Results and Discussion (Part 2).
The roots and fruits of Ferula drudeana were analyzed, and details of the preparation of essential oil and solvent extracts were described in sections 2.1 and 3.3 of the revised manuscript.
Specific comments:
1. In abstract clear aim of the study should be presented in first line. Please rearrange text after including the aim.
As per the reviewer's request, the abstract was revised to include the aim of the study.
2. Lines 50-53, please verify if defined aim reflected to the study content.
The defined aim is reflected in the relevant section of the revised manuscript.
3. Page 3 , I understand that in in Lines 61-64 explanation to the Table 1 is presented. Thus I suggest to decrease font size this description and put under the Table. Methodology of identification compounds should be moved to the appropriate part 3.
The font size is reduced in the footnote section of Table 1 and methodology of the identification of compounds is moved to the appropriate section of part 3 in the revised manuscript.
4. Figure 1: please insert names of all compounds in caption of Figure 1…..; 2…….
In the structural formula figures, the names of the compounds were included along with the corresponding structures.
5. Part 2.1, the discussion of volatile composition of the fruit essential oil is not sufficiently described, please supply more information, for example, about properties of more abundant compounds.
Information about the properties of more abundant compounds was provided in the last part of section 2.1.
6. The part 3.7. should be supplemented with information about qualitative and quantitative analysis of compounds.
Section 3.7. is supplemented with qualitative and quantitative analysis of extracts and compounds (standards) in the revised manuscript.
7. Also in part 3 please insert detailed information about of equipment (producer, city, country).
The equipment information is included in section 3.1 of the revised manuscript.
8. Authors analysed methanolic extract, that is necessary to include some information about limitation of using this extract (as ingredient in supplements formula).
As explained in section 3.3 of the revised manuscript, the extraction solvent of each extract was completely removed in vacuo to yield the final solvent-free extract. Nevertheless, none of the extracts used in this study was intended to be used as ingredient in supplements.
Thank you very much for your invaluable comments.
Kind regards,
M. Miski, Ph.D.
Round 2
Reviewer 1 Report
The suggestions I asked for in an earlier report have been taken into account. The method with the DPPH radical in the revised form is familiar to me and is understandable to me. Keep up the good work, good luck with your further research.
Reviewer 3 Report
The manuscript was sufficiently revised by the Authors. I also appreciate responses on all comments.